# Effect of hearing aids on body balance function in non-reverberant condition: A posturographic study

Chihiro Ninomiya[1], Harukazu Hiraumi[1]*, Kiyoshi Yonemoto[2], Hiroaki Sato[1]

1 Department of Otolaryngology—Head and Neck Surgery, Iwate Medical University, Shiwa, Iwate, Japan,
2 Faculty of Social Welfare, Iwate Prefectural University, Takizawa, Iwate, Japan

* hhiraumi@iwate-med.ac.jp

**Data Availability Statement:** All relevant data are within the manuscript.

**Funding:** This work was supported by JSPS KAKENHI (HH, grant number JP17K11341). The

## Abstract

### Objective

The purpose of this study was to evaluate the effect of hearing aids on body balance function in a strictly controlled auditory environment.

### Methods

We recorded the findings of 10 experienced hearing aid users and 10 normal-hearing participants. All the participants were assessed using posturography under eight conditions in an acoustically shielded non-reverberant room: (1) eyes open with sound stimuli, with and without foam rubber, (2) eyes closed with sound stimuli, with and without foam rubber, (3) eyes open without sound stimuli, with and without foam rubber, and (4) eyes closed without sound stimuli, with and without foam rubber.

### Results

The auditory cue improved the total path area and sway velocity in both the hearing aid users and normal-hearing participants. The analysis of variance showed that the interaction among eye condition, sound condition, and between-group factor was significant in the maximum displacement of the center-of-pressure in the mediolateral axis (F [1, 18] = 6.19, $p$ = 0.02). The maximum displacement of the center-of-pressure in the mediolateral axis improved with the auditory cues in the normal-hearing participants in the eyes closed condition (5.4 cm and 4.7 cm, $p$ < 0.01). In the hearing aid users, this difference was not significant (5.9 cm and 5.7 cm, $p$ = 0.45). The maximum displacement of the center-of-pressure in the anteroposterior axis improved in both the hearing aid users and the normal-hearing participants.

## 1. Introduction

Body balance is maintained by multiple sensory systems, including the vestibular, visual, and proprioceptive systems. These systems deteriorate with age, resulting in persistent dizziness or

funders had no role in study design, data collection and analysis, decision to publish, or preparation of the manuscript.

**Competing interests:** The authors have declared that no competing interests exist.

instability. Balance dysfunction is a risk factor for falls, and is closely associated with multifaceted symptoms of frailty. Despite the high frequency and clinical importance of impaired balance functions, they are imperceptible to others and are easily underestimated. In addition, balance deficiency has no standard treatment besides rehabilitation.

Recently, hearing compensation has been reported to improve the balance function; however, the benefits of hearing compensation on the balance function are still controversial [1,2]. Rumalla et al. performed Romberg and Mann tests on sponges in the elderly and reported that hearing aids (HAs) improved them [3]. Vitkovic et al. evaluated the balance function using static posturography [4]. In their study, they showed that the sound environment affects the mean path lengths differently in aided and unaided conditions. In addition, the vestibular impaired participants had higher path lengths compared to the participants with normal balance, and the difference increased in the absence of sound. Negahban et al. reported that HAs improved the standard deviation velocity within the eyes open-foam surface condition, whereas the total path area was not affected by the sound condition [5]. Maheu et al. reported that hearing loss participants with vestibular loss benefited significantly more from HAs in sway area compared to normal-hearing and hearing impaired participants with normal vestibular function on foam platform [6]. This difference was not observed in the analysis of the sway velocities. In contrast to these studies, McDaniel et al. failed to find any variation in the participants' balance regardless of the presence or absence of their HAs during the Sensory Organization Test [7].

One of the reasons for this discrepancy is the uncontrolled sound conditions in these studies. Auditory space perception is thought to contribute the body balance improvement. Auditory space perception is influenced by multiple factors, including the spectrum and azimuth. In addition, the echoic sound disrupts the auditory space recognition, which suggests that the control of reverberation decreases the heterogeneity of the effect of auditory cues on body balance. Only a limited number of studies have reported the reverberation properties of the examination environment [8,9]. Precise sound control is required to evaluate the actual influence of hearing compensation on the body balance.

In this study, we investigated the body balance function of HA users in an anechoic, sound-shielded room using posturography to clarify the effect of hearing compensation.

## 2. Methods

### 2.1 Participants

We evaluated 10 hearing-loss patients with HAs on both sides. They were recruited at the Iwate Medical University (5 males, 5 females; aged 62–79 years, mean age 63.4 years). All the participants had been using HAs for more than 1 year (1–8 years, mean 3.7 years). None of them showed abnormalities in vestibular test including vestibulo-ocular reflex test and head impulse test. Ten normal-hearing paid volunteers (10 females; aged between 62 and 75 years, mean 71.2 years) with no history of neurological or muscular diseases and who showed normal hearing threshold were recruited as controls.

All participants provided written informed consent, and the study protocol was approved by the Ethics Committee of Iwate Medical University (MH2019-158), in accordance with the Declaration of Helsinki.

### 2.2 Postural sway measurement

All the participants were assessed using posturography (GP-5000, Anima Co., Ltd., Tokyo, Japan) in an anechoic room. All the participants were required to stand on a flat platform with their feet close together for 60 s. The medial surfaces of the toes and calcanei were aligned to the centerline of the platform. HA users wore their own HA during the experiment. The force

transducers embedded in the platform continuously measured the displacement of the center-of-foot pressure (COP) with a sampling frequency of 20 Hz. The COP is approximately the same as the position of the center of gravity while standing still.

In the HA users, the posturographic measurements were conducted under eight conditions: (1) eyes open on a foam pad, with and without HA, (2) eyes closed on a foam pad, with and without HA, (3) eyes open on a rigid surface, with and without HA, and (4) eyes closed on a rigid surface, with and without HA. The normal-hearing participants underwent a similar measurement with and without sound, instead of with and without HAs. The sequence of the eight conditions was counterbalanced among the participants. In conditions with sound stimuli, white noise (70 dBA at the position of the head center of each participant) was delivered from a loud speaker (101VM, BOSE, Massachusetts, USA, frequency range 70–17,000 Hz: IEC60581-7) placed 1 m anterior to the participants at their ear level. The position of the speaker was adjusted using a laser level (AL-50V, OHTA manufactory, Tokyo, Japan) and a laser rangefinder (LS-411, MAX Co., Ltd., Tokyo, Japan). All the measurements were conducted in an anechoic room constructed by the Wakabayashi Acoustic Design Corporation (Tokyo, Japan). The size of the room was 5,400 mm (width) x 4,800 mm (length) x 3,000 mm (height). The ambient noise level was less than 15 dBA between 125 and 16,000 Hz. At the center of this room, the free-field decay of sound from a point source was verified to follow the inverse square law between 250 and 8,000 Hz. These were measured using a microphone, a preamplifier, and a measuring amplifier, calibrated with an acoustic calibrator (Type 4190, Type 2669, Type 2636, and Type 4226, respectively, Bruel&Kjaer, Naerum, Denmark).

### 2.3 Statistical analysis

The total path area and the average sway velocity were examined by three-way repeated measures and mixed factorial analyses of variance (ANOVA) with a within-group factor of eye conditions (eyes open/eyes closed), sound conditions (with sound/without sound), and a foam condition (with foam/without foam), and a between-group factor (HA users/normal-hearing participants). The total path area and the averaged velocity were further analyzed in the mediolateral (ML) and anteroposterior (AP) axes using the maximum displacement of the COP and the averaged velocity in the ML and AP axes. In addition, the same analysis was conducted in the HA users only using three-way repeated measures ANOVA to clarify the effect of HAs.

Post-hoc pairwise comparison with Bonferroni adjustment was conducted when the interaction among the conditions was significant. Statistical significance was set at $p < 0.05$. All the analyses were conducted using SPSS software (IBM SPSS Statistics 24 for Windows, Advanced Analytics Inc., Tokyo, Japan).

## 3. Results

The demographic data of the participants are summarized in Table 1. No history of neurological or muscular diseases other than hearing loss was reported. None of the participants required assistance in preventing a fall during the experiment under the eight conditions described above. No adverse effects were observed before, during, or after the experiment. All the parameters were obtained for all the participants. The results are presented in Table 2.

### 3.1. Analysis of the total area

The ANOVA showed a significant interaction between the sound condition and foam condition ($F\ [1, 18] = 7.04$, $p = 0.02$), between sound condition and eye condition ($F\ [1, 18] = 5.77$, $p = 0.03$), and between eye and foam conditions ($F\ [1, 18] = 96.66$, $p < 0.01$). The pair-wise post-hoc analysis revealed that the sound significantly decreased the total area in the condition

**Table 1. The background of the hearing aid users.**

| No | Age | Sex | Duration of HA use | Average of hearing |
|----|-----|-----|--------------------|--------------------|
| 1 | 62 | M | 8years | 65.0 dBHL |
| 2 | 76 | M | 2years | 52.0 dBHL |
| 3 | 77 | M | 2years | 60.0 dBHL |
| 4 | 64 | M | 1years | 63.8 dBHL |
| 5 | 73 | F | 7years | 70.0 dBHL |
| 6 | 69 | F | 2years | 60.0 dBHL |
| 7 | 72 | M | 5years | 65.0 dBHL |
| 8 | 80 | F | 8years | 63.8 dBHL |
| 9 | 62 | F | 1years | 53.8 dBHL |
| 10 | 64 | F | 7years | 57.5 dBHL |

HA: Hearing aids, M: Male, F: Female.

with foam (15.54 cm$^2$ and 13.58 cm$^2$, $p < 0.01$) and in the condition with eyes closed (15.37 cm$^2$ and 13.43 cm$^2$, $p < 0.01$). There were no significant differences between the HA users and normal-hearing participants.

The analysis in HA users showed that the HAs decreased the total path area (10.95 cm$^2$ and 9.85 cm$^2$, F [1, 9] = 5.74, $p = 0.04$).

## 3.2. Analysis of the maximum COP displacement in the mediolateral axis

The ANOVA showed that the interaction among the eye condition, sound condition, and between-group factor was significant (F [1, 18] = 6.19, $p = 0.02$). The pair-wise post-hoc analysis revealed that the sound decreased the maximum COP displacement in the ML axis of the normal-hearing participants in eyes closed condition (5.44 cm and 4.67 cm, $p < 0.01$). In HA users, this difference was not significant (5.86 cm and 5.73 cm, $p = 0.45$) (Fig 1A and 1B). In the eyes open condition, the sound did not change the maximum COP displacement in the ML axis (2.57 cm and 2.37 cm, $p = 0.13$ in normal-hearing participants, 2.77 cm and 2.66 cm, $p = 0.36$ in HA users). The interaction between foam condition and eye condition was also significant (F (1, 18) = 167.32, $p < 0.01$).

The analysis in HA users showed that the HAs did not change the maximum COP displacement in the ML axis (4.32 cm and 4.19 cm, F (1, 9) = 0.88, $p = 0.37$).

In hearing aid users, the maximum displacement of the center of pressure did not change in the with and without sound conditions (Fig 1A). In normal-hearing participants, the maximum displacement of the center of pressure decreased in the with sound condition (Fig 1B). The differences were statistically significant ($p < 0.01$).

The COP is the center of pressure. The dashed line indicates a condition with a foam surface, and the rigid line indicates a condition with a rigid surface.

## 3.3. Analysis of the maximum COP displacement in the anteroposterior axis

The ANOVA showed a significant main effect of sound condition (F (1, 18) = 7.95, $p = 0.01$) and a significant interaction between eye condition and foam condition (F (1, 18) = 70.88, $p < 0.01$). The pair-wise post-hoc analysis revealed that the sound decreased the maximum COP displacement in the AP axis (4.40 cm and 4.18 cm, $p = 0.01$). There were no significant differences between the HA users and normal-hearing participants.

The analysis in HA users showed that the HAs decreased the maximum COP displacement in the AP axis (4.53 cm and 4.26 cm, F (1, 9) = 9.86, $p = 0.01$).

**Table 2. Results of static posturography in hearing aid users and normal-hearing participants.**

| | HA users | | NH participants | |
| --- | --- | --- | --- | --- |
| | Mean (SEM) | | Mean (SEM) | |
| | Sound(+) | Sound(-) | Sound(+) | Sound(-) |
| Eyes open-Rigid surface | | | | |
| Total Area | 3.04(0.87) | 2.98(0.96) | 2.21(0.30) | 1.97(0.27) |
| Max displacement, ML | 2.25(0.33) | 2.31(0.32) | 1.98(0.18) | 1.88(0.17) |
| Max displacement, AP | 2.80(0.40) | 2.79(0.52) | 2.65(0.23) | 2.41(0.16) |
| Sway Velocity | 1.48(0.23) | 1.73(0.30) | 1.11(0.07) | 1.11(0.08) |
| Velocity, ML | 0.80(0.11) | 0.94(0.14) | 0.62(0.05) | 0.64(0.05) |
| Velocity, AP | 0.80(0.15) | 0.94(0.20) | 0.62(0.05) | 0.61(0.05) |
| Eyes open-Foam surface | | | | |
| Total Area | 5.41(1.44) | 5.76(1.35) | 4.79(0.66) | 5.56(0.63) |
| Max displacement, ML | 3.06(0.39) | 3.24(0.50) | 2.78(0.16) | 3.27(0.19) |
| Max displacement, AP | 3.76(0.40) | 3.78(0.31) | 3.69(0.36) | 4.34(0.34) |
| Sway Velocity | 2.29(0.35) | 2.35(0.25) | 2.08(0.11) | 2.21(0.15) |
| Velocity, ML | 1.21(0.17) | 1.27(0.14) | 1.12(0.05) | 1.18(0.06) |
| Velocity, AP | 1.22(0.20) | 1.25(0.08) | 1.19(0.07) | 1.28(0.11) |
| Eyes closed-Rigid surface | | | | |
| Total Area | 5.48(1.04) | 6.11(1.20) | 4.09(0.69) | 4.52(0.43) |
| Max displacement, ML | 3.62(0.36) | 3.59(0.42) | 2.80(0.29) | 3.40(0.20) |
| Max displacement, AP | 3.46(0.35) | 3.66(0.37) | 3.21(0.32) | 3.32(0.22) |
| Sway Velocity | 2.43(0.24) | 2.55(0.20) | 1.80(0.16) | 2.01(0.19) |
| Velocity, ML | 1.35(0.14) | 1.43(0.15) | 1.00(0.09) | 1.23(0.12) |
| Velocity, AP | 1.31(0.16) | 1.36(0.10) | 1.02(0.11) | 1.01(0.10) |
| Eyes closed-Foam surface | | | | |
| Total Area | 25.47(3.73) | 28.94(3.58) | 18.66(1.65) | 21.90(1.61) |
| Max displacement, ML | 7.83(0.49) | 8.12(0.60) | 6.54(0.41) | 7.48(0.33) |
| Max displacement, AP | 7.01(0.50) | 7.88(0.58) | 6.83(0.33) | 6.99(0.54) |
| Sway Velocity | 6.52(0.80) | 6.93(0.80) | 5.10(0.55) | 5.27(0.30) |
| Velocity, ML | 3.32(0.42) | 3.61(0.43) | 2.91(0.33) | 2.92(0.14) |
| Velocity, AP | 3.51(0.38) | 3.66(0.37) | 2.80(0.18) | 2.95(.019) |

HA: Hearing aid, NH: Normal-hearing, SEM: Standard error of mean.

AP: Anteroposterior, ML: Mediolateral.

Units of parameters are as follows: cm$^2$ (area), cm (displacement), cm/s (velocity).

### 3.4. Analysis of the averaged sway velocity

The ANOVA showed a significant main effect of sound condition (F (1, 18) = 9.06, $p < 0.01$) and a significant interaction between the eye condition and foam condition (F (1, 18) = 74.75, $p < 0.01$). The pairwise post-hoc analysis revealed that the sound decreased the average sway velocity (3.02 cm/s and 2.85 cm/s, $p < 0.01$). There were no significant differences between the HA users and normal-hearing participants.

The analysis in HA users showed that the HAs decreased the averaged sway velocity (3.39 cm/s and 3.18 cm/s, F (1, 9) = 8.51, $p = 0.02$).

### 3.5. Analysis of the averaged sway velocity in the mediolateral axis

The ANOVA showed a significant main effect of sound condition (F (1, 18) = 6.53, $p = 0.02$) and a significant interaction between the eye condition and foam condition (F (1, 18) = 66.08,

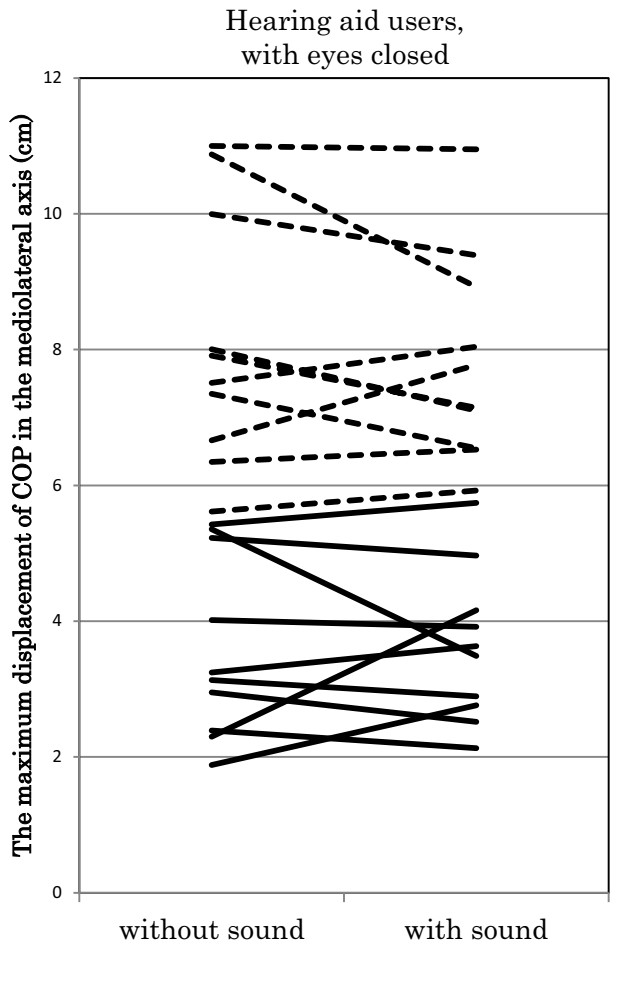

Hearing aid users,
with eyes closed

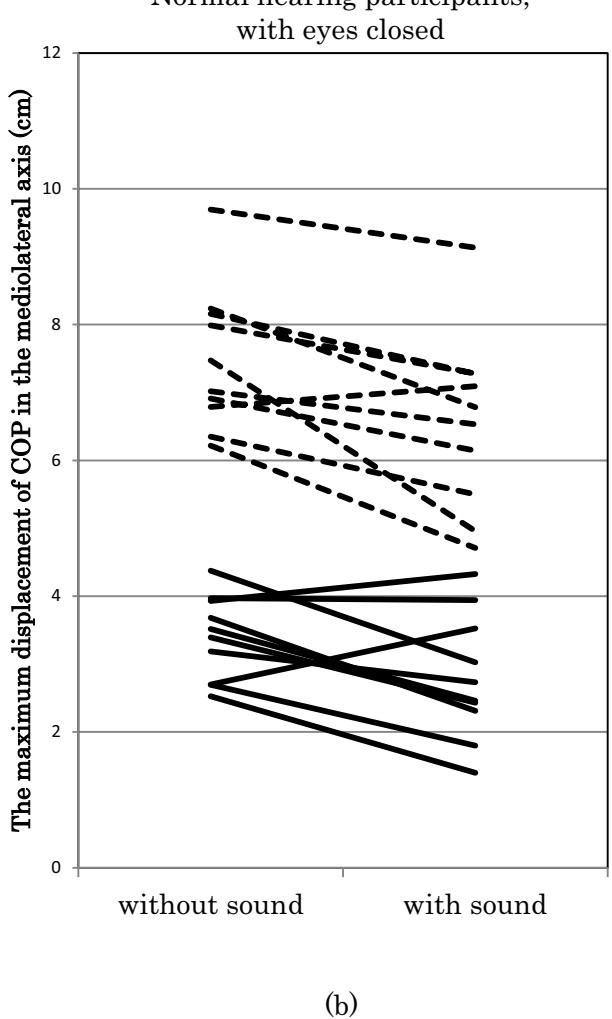

Normal hearing participants,
with eyes closed

(a)

(b)

**Fig 1. The maximum displacement of the center of pressure in the mediolateral direction with eyes closed.**

$p < 0.01$). The pair-wise post-hoc analysis revealed that the sound decreased the averaged sway velocity in the ML axis (1.65 cm/s and 1.54 cm/s, $p = 0.02$). There were no significant differences between the HA users and normal-hearing participants.

The analysis in the HA users showed that the HAs decreased the averaged sway velocity in the ML axis (1.81 cm/s and 1.67 cm/s, F [1, 9] = 6.93, $p = 0.03$) (Fig 2).

In the hearing aid users, the average sway velocity in the mediolateral direction decreased in the group with sound condition. This change was statistically significant.

The dashed line indicates a condition with a foam surface, and the rigid line indicates a condition with a rigid surface.

### 3.6. Analysis of the averaged sway velocity in the anteroposterior axis

The ANOVA showed a significant main effect of sound condition (F (1, 18) = 10.44, P < 0.01) and a significant interaction between the eye condition and foam condition (F (1, 18) = 94.10, P < 0.01). The pair-wise post-hoc analysis revealed that the sound decreased the average sway

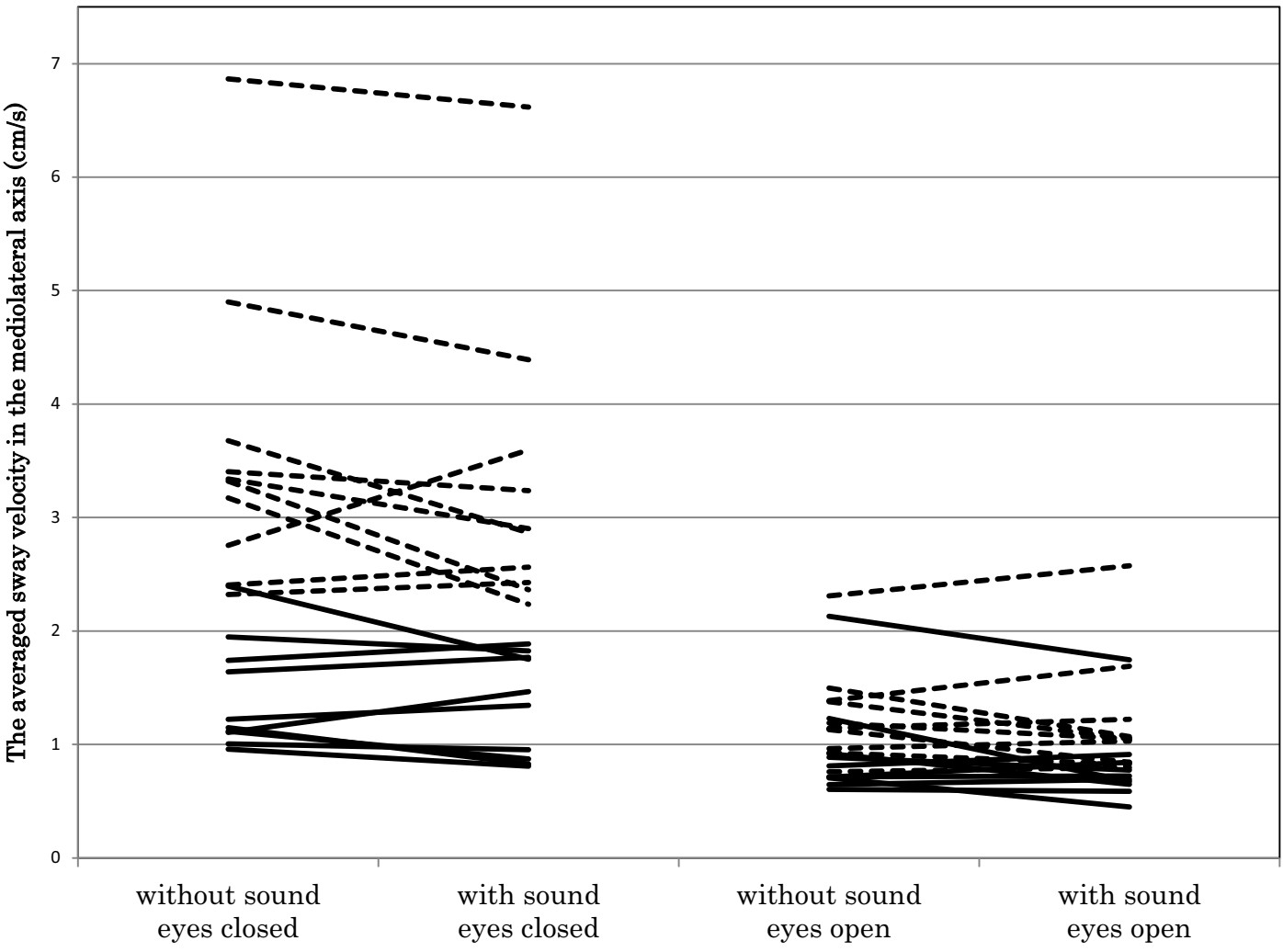

**Fig 2. The averaged sway velocity in the mediolateral direction in hearing aid users.**

velocity in the AP axis (1.63 cm/s and 1.56 cm/s, $p < 0.01$). There were no significant differences between the HA users and normal-hearing participants.

The analysis in the HA users showed that the HAs decreased the averaged sway velocity in the AP axis (1.80 cm/s and 1.71 cm/s, $F[1, 9] = 10.42$, $p = 0.01$).

## 4. Discussion

In this study, we demonstrated that auditory cues improved the total path area in challenging conditions in both HA users and normal-hearing participants. In HA users, this improvement was observed only in the AP axis. The averaged velocity improved with auditory cues in both the HA users and the normal-hearing participants.

Previous studies have hypothesized that the mechanism of body balance improvement by auditory cues is due to auditory space recognition. This hypothesis implies that the COP

displacement in the ML axis depends on the minimum audible movement angle, and the COP displacement in the AP axis depends on the minimum audible movement distance. The minimum audible movement angle and the minimum audible movement distance are influenced by various factors, including differences in sound, duration of the stimuli, reverberant sound, velocity of the source and listener, and so on. The minimum audible movement angle has been explored in various settings; in the condition with broadband noise and with low velocity setting, similar to the present study, the minimum audible movement angle in the horizontal plane is approximately 1.5 degrees [10], corresponding to 5.2 cm in the ML axis in the setting of the present study. In elderly participants with hearing impairment, the minimum audible movement angle is reported to be large [11]. This could explain the findings of our present study that the maximum COP displacement in the ML axis was unaffected by the sound presentation.

The minimum audible movement distance perception has rarely been explored. Instead of minimum audible movement distance perception, auditory distance perception has been investigated in much more detail [11]. The overall sound level and the direct-to-reverberant ratio are the two major cues for auditory distance perception [12]. The interaural time and level differences can be cues for distances closer than approximately 1 m [13]. Spectral shape and the motion of sound source also provide auditory distance information [14]. The contribution of these cues to the auditory distance perception varies according to the experiment condition. The effect of direct-to-reverberant ratio are attenuated in an anechoic environment. The binaural cues are minimum in frontal sound source. The spectral and dynamic cues benefit the auditory distance perception for very close or distant sound source [14]. In the present experiment, we presented broadband noise from a loud speaker placed 1 m anterior to the participants in the medial plane in the anechoic chamber; hence, the overall sound level was the most important cue for auditory distance perception. There are relatively few studies on the effect of hearing loss on auditory distance perception. Hearing loss adversely affects the use of direct-to-reverberant ratio cues; however, the use of level cues remains relatively unaffected [14,15]. This may explain why the sound presentation improved the maximum COP displacement in the AP axis in both participants with normal hearing and those with hearing loss.

In addition to the total path area, the sway velocity improved after sound presentation in both normal-hearing participants and HA users. Previous studies reported similar results wherein the auditory cue decreased the total path length or sway velocity [4,5]. These previous studies attributed the results to the improved auditory space perception. In the present study, however, the maximum displacement of the COP in the ML axis was not improved by sound presentation in the HA users. This suggests that the auditory cue decreased the sway velocity through a mechanism different from auditory space perception. Stimulation of the cochlea with intense sound is known to stimulate the vestibular system [9,16], regardless of hearing function. Recently, constant stimulation of the otolith organ has been reported to improve body balance function measured using posturography [17]. It is possible that the amplified sound stimulated the otolith organ constantly, which resulted in improved sensitivity to gravity and reduced sway velocity in the present study.

This study has a few limitations. First, the study enrolled a comparatively small number of participants. A limited number of participants was enrolled since the safety of the anechoic room for HA users was not clearly established. Nevertheless, we found a beneficial effect of auditory cues thorough HAs on body stabilization under static conditions, which may justify our hypothesis that the controlled sound environment decreases the heterogeneity of the effect of auditory cues on body balance. Second, the normal participants in the present study were all female. We recruited normal-hearing paid volunteers regardless of the gender. Most of the attendee were female, and none of the male attendees met the inclusion criteria. This may be

due to the high prevalence of age-related hearing loss in men [18]. Recent posturographic study showed that the gender difference was not statistically significant [19]. Despite that, females are known to be more prone to falls than males. It may be preferable to recruit gender-matched control participants in further studies. Third, the auditory environment used in this study was not identical to the normal sound environment in daily life. All the participants were of normal vestibular function. Further studies exploring dynamic and reverberant conditions and vestibulopathy participants are warranted.

## 5. Conclusion

Auditory cues improved the body sway area and velocity in both HA users and normal-hearing participants in an anechoic chamber. In the HA users, the maximum COP displacement in the ML axis was not affected by sound stimulation, whereas the sway velocity in the ML axis improved with sound stimulation. This suggests that the underlying mechanism for the improvement of these two parameters is different. Further research is warranted to prove the effectiveness of auditory information when wearing HAs in real life.

## Acknowledgments

We would like to thank Editage (www.editage.com) for English language editing.

## Author Contributions

**Conceptualization:** Harukazu Hiraumi.

**Data curation:** Harukazu Hiraumi, Kiyoshi Yonemoto, Hiroaki Sato.

**Formal analysis:** Chihiro Ninomiya, Harukazu Hiraumi, Kiyoshi Yonemoto.

**Funding acquisition:** Harukazu Hiraumi.

**Investigation:** Chihiro Ninomiya, Harukazu Hiraumi, Hiroaki Sato.

**Methodology:** Chihiro Ninomiya, Harukazu Hiraumi, Kiyoshi Yonemoto, Hiroaki Sato.

**Project administration:** Harukazu Hiraumi, Hiroaki Sato.

**Supervision:** Harukazu Hiraumi, Kiyoshi Yonemoto, Hiroaki Sato.

**Validation:** Chihiro Ninomiya, Harukazu Hiraumi, Kiyoshi Yonemoto, Hiroaki Sato.

**Visualization:** Chihiro Ninomiya, Harukazu Hiraumi.

**Writing – original draft:** Chihiro Ninomiya.

**Writing – review & editing:** Harukazu Hiraumi, Kiyoshi Yonemoto, Hiroaki Sato.

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
