## [Decision Letter · Decision Letter 0]

23 Aug 2021

PONE-D-21-23190

Effect of hearing aids on body balance function in non-reverberant condition: A posturographic study

PLOS ONE

Dear Dr. Hiraumi,

Thank you for submitting your manuscript to PLOS ONE. After careful consideration, we feel that it has merit but does not fully meet PLOS ONE’s publication criteria as it currently stands. Therefore, we invite you to submit a revised version of the manuscript that addresses the points raised during the review process.

This Academic Editor has to express my apology to be late for this decision letter, because the third reviewer has not yet given the comments within a due date. 

Two experts in the field have carefully reviewed the manuscript entitled, "Effect of hearing aids on body balance function in non-reverberant condition: A posturographic study". Their comments are appended below.

Both reviewers appreciate the manuscript with leaving some minor concerns which should be clarified before publication.

I will expect to receive the replies to each critiques and necessary revision.

We look forward to receiving your revised manuscript.

Kind regards,

Manabu Sakakibara, Ph.D.

Academic Editor

PLOS ONE

Journal Requirements:

 [This work was supported by JSPS KAKENHI (HH, grant number JP17K11341).]  

[This work was supported by JSPS KAKENHI (grant number JP17K11341). We would like to thank Editage (www.editage.com) for English language editing.]

 [This work was supported by JSPS KAKENHI (HH, grant number JP17K11341).]

[No authors have competing interests]. 

Reviewers' comments:

Reviewer's Responses to Questions

**Comments to the Author**

1. Is the manuscript technically sound, and do the data support the conclusions?

Reviewer #1: Yes

Reviewer #2: Yes

2. Has the statistical analysis been performed appropriately and rigorously? 

Reviewer #1: I Don't Know

Reviewer #2: Yes

3. Have the authors made all data underlying the findings in their manuscript fully available?

Reviewer #1: Yes

Reviewer #2: Yes

4. Is the manuscript presented in an intelligible fashion and written in standard English?

Reviewer #1: Yes

Reviewer #2: Yes

5. Review Comments to the Author

Reviewer #1: The study is interesting but it has some limitations such as small number of participants that authors mentioned. All of the normal hearing cases were female. why did the authors choose just females for normal hearing group?

in table 2, its better to add "p values" in the table. Although the authors found significant difference in posturography values but they did not evaluate the balance status by other tests to show its "clinical importance". The figures are difficult to interpret. By the way the study is interesting for me.

Reviewer #2: Please give more details for auditory distance perception , make sure that no other dual publication for this paper not occurred , you must follow the research ethical rules for privacy and honesty for patients

6. PLOS authors have the option to publish the peer review history of their article (what does this mean?). If published, this will include your full peer review and any attached files.

Reviewer #1: **Yes: **Alimohamad Asghari

Reviewer #2: **Yes: **Salwa mahmoud

---

## [Author Response · Author response to Decision Letter 0]

23 Sep 2021

Dear Professor Alimohamad Asghari and Professor Salwa Mahmoud,

Thank you for positive and constructive comments. We revised the paper according to your recommendation.

Reviewer #1: The study is interesting but it has some limitations such as small number of participants that authors mentioned. 

1) All of the normal hearing cases were female. why did the authors choose just females for normal hearing group?

Answer.

Thank you for the suggestion. We recruited normal participants regardless of the gender. Most of the attendee were unexpectedly female. In addition, all the attended males did not fulfil our inclusion criteria (mainly high pure tone threshold). 

The following descriptions were added.

L288

“Second, the normal participants in the present study were all female. We recruited normal-hearing paid volunteers regardless of the gender. Most of the attendee were female, and none of the male attendees met the inclusion criteria. This may be due to the high prevalence of age-related hearing loss in men (18). Recent posturographic study showed that the gender difference was statistically significant (19). Despite that, females are known to be more prone to falls than males. It may be preferable to recruit gender-matched control participants in further studies.”

2) In table 2, it is better to add "p values" in the table.

Answer.

We made the table according to the previous report (ref #5, Negahban et al., 2017). As the reviewer #1 suggested, it is better to include the result of statistics in the table. The problem is that we used three-way repeated measures and mixed factorial ANOVA. We like this procedure, but it is not easy to include the results of ANOVA in the table. 

The following is a reformatted table that include the results of ANOVA. This table is informative but I feel it too busy. I like the original table, but am ready to use the reformatted table if the reviewer like the new one. 

 HA users NH participants

 Mean (SEM) Mean (SEM)

 Sound(+) Sound(-) Sound(+) Sound(-)

Total Area

Significant interaction between the sound condition and foam condition (F [1, 18] =7.04, p = 0.02), between sound condition and eye condition (F [1, 18] = 5.77, p = 0.03), and between eye and foam conditions (F [1, 18] = 96.66, p < 0.01)

 Eyes open-Rigid surface 3.04(0.87) 2.98(0.96) 2.21(0.30) 1.97(0.27)

 Eyes open-Foam surface 5.41(1.44) 5.76(1.35) 4.79(0.66) 5.56(0.63)

 Eyes closed-Rigid surface 5.48(1.04) 6.11(1.20) 4.09(0.69) 4.52(0.43)

 Eyes closed-Foam surface 25.47(3.73) 28.94(3.58) 18.66(1.65) 21.90(1.61)

Maximum displacement, ML

Significant interaction among the eye condition, sound condition, and between-group factor (F [1, 18] = 6.19, p = 0.02)

 Eyes open-Rigid surface 2.25(0.33) 2.31(0.32) 1.98(0.18) 1.88(0.17)

 Eyes open-Foam surface 3.06(0.39) 3.24(0.50) 2.78(0.16) 3.27(0.19)

 Eyes closed-Rigid surface 3.62(0.36) 3.59(0.42) 2.80(0.29) 3.40(0.20)

 Eyes closed-Foam surface 7.83(0.49) 8.12(0.60) 6.54(0.41) 7.48(0.33)

Maximum displacement, AP

Significant main effect of sound condition (F (1, 18) = 7.95, p = 0.01) and a significant interaction between eye condition and foam condition (F (1, 18) =70.88, p < 0.01)

 Eyes open-Rigid surface 2.80(0.40) 2.79(0.52) 2.65(0.23) 2.41(0.16)

 Eyes open-Foam surface 3.76(0.40) 3.78(0.31) 3.69(0.36) 4.34(0.34)

 Eyes closed-Rigid surface 3.46(0.35) 3.66(0.37) 3.21(0.32) 3.32(0.22)

 Eyes closed-Foam surface 7.01(0.50) 7.88(0.58) 6.83(0.33) 6.99(0.54)

Sway Velocity

Significant main effect of sound condition (F (1, 18) = 9.06, p < 0.01) and a significant interaction between the eye condition and foam condition (F (1, 18) =74.75, p < 0.01)

 Eyes open-Rigid surface 1.48(0.23) 1.73(0.30) 1.11(0.07) 1.11(0.08)

 Eyes open-Foam surface 2.29(0.35) 2.35(0.25) 2.08(0.11) 2.21(0.15)

 Eyes closed-Rigid surface 2.43(0.24) 2.55(0.20) 1.80(0.16) 2.01(0.19)

 Eyes closed-Foam surface 6.52(0.80) 6.93(0.80) 5.10(0.55) 5.27(0.30)

Velocity, ML

Significant main effect of sound condition (F (1, 18) = 6.53, p = 0.02) and a significant interaction between the eye condition and foam condition (F (1, 18) =66.08, p < 0.01)

 Eyes open-Rigid surface 0.80(0.11) 0.94(0.14) 0.62(0.05) 0.64(0.05)

 Eyes open-Foam surface 1.21(0.17) 1.27(0.14) 1.12(0.05) 1.18(0.06)

 Eyes closed-Rigid surface 1.35(0.14) 1.43(0.15) 1.00(0.09) 1.23(0.12)

 Eyes closed-Foam surface 3.32(0.42) 3.61(0.43) 2.91(0.33) 2.92(0.14)

Velocity, AP

Significant main effect of sound condition (F (1, 18) = 10.44, P < 0.01) and a significant interaction between the eye condition and foam condition (F (1, 18) =94.10, P < 0.01)

 Eyes open-Rigid surface 0.80(0.15) 0.94(0.20) 0.62(0.05) 0.61(0.05)

 Eyes open-Foam surface 1.22(0.20) 1.25(0.08) 1.19(0.07) 1.28(0.11)

 Eyes closed-Rigid surface 1.31(0.16) 1.36(0.10) 1.02(0.11) 1.01(0.10)

 Eyes closed-Foam surface 3.51(0.38) 3.66(0.37) 2.80(0.18) 2.95(.019)

3) Although the authors found significant difference in posturography values but they did not evaluate the balance status by other tests to show its "clinical importance". 

Answer.

Thank you for the advice. 

I agree with the referee that it is better to include participants with vestibulopathy. 

We tested the participants with vestibulo-ocular reflex test and head impulse test to exclude severe vestibulopathy, as was conducted in the previous study (ref #4, Vitkovic et al., 2016). However, we found that participants with vestibulopathy cannot complete the experiment with foam surface (ref #9, Oikawa et al., 2018). Based on this finding, we conducted the present study only in those with normal vestibular functions.

The following descriptions were added.

L79

“None of them showed abnormalities in vestibular test including vestibulo-ocular reflex test and head impulse test.”

L294

“All the participants were of normal vestibular function. Further studies exploring dynamic and reverberant conditions and vestibulopathy participants are warranted.”

4) The figures are difficult to interpret. 

Answer.

In this study, we used three-way repeated measures and mixed factorial ANOVA. The figures were made to represent the results of the statistics. The figures can be made simple if we use the average and SEM, but it will reduce the individual information. We made the figures so as not to delete the raw data. In addition, we can compare the velocity and displacement in hearing aid users with eyes closed (Fig 1(a) and left column of Fig 2).

I will appreciate it if the reviewers accept the figures in the present form.

 

Reviewer #2: 

1) Please give more details for auditory distance perception 

Answer.

Thank you for the advice. We add more detail about the auditory distance perception.

L286

“Second, the normal participants in the present study were all female. We recruited normal-hearing paid volunteers regardless of the gender. Most of the attendee were female, and none of the male attendees met the inclusion criteria. This may be due to the high prevalence of age-related hearing loss in men (18). Recent posturographic study showed that the gender difference was not statistically significant (19). Despite that, females are known to be more prone to falls than males. It may be preferable to recruit gender-matched control participants in further studies.”

2) Make sure that no other dual publication for this paper not occurred

Answer.

I confirm that no other dual publication for this paper occurred.

3) You must follow the research ethical rules for privacy and honesty for patients

Answer.

This study was conducted in accordance with the Declaration of Helsinki. All the identifying information were removed from the analyzed data.

---

## [Decision Letter · Decision Letter 1]

1 Oct 2021

Effect of hearing aids on body balance function in non-reverberant condition: A posturographic study

PONE-D-21-23190R1

Dear Dr. Hiraumi,

We’re pleased to inform you that your manuscript has been judged scientifically suitable for publication and will be formally accepted for publication once it meets all outstanding technical requirements.

Kind regards,

Manabu Sakakibara, Ph.D.

Academic Editor

PLOS ONE

Additional Editor Comments (optional):

Reviewers' comments:

Reviewer's Responses to Questions

**Comments to the Author**

1. If the authors have adequately addressed your comments raised in a previous round of review and you feel that this manuscript is now acceptable for publication, you may indicate that here to bypass the “Comments to the Author” section, enter your conflict of interest statement in the “Confidential to Editor” section, and submit your "Accept" recommendation.

Reviewer #1: All comments have been addressed

Reviewer #2: All comments have been addressed

2. Is the manuscript technically sound, and do the data support the conclusions?

Reviewer #1: Yes

Reviewer #2: Yes

3. Has the statistical analysis been performed appropriately and rigorously? 

Reviewer #1: Yes

Reviewer #2: Yes

4. Have the authors made all data underlying the findings in their manuscript fully available?

Reviewer #1: Yes

Reviewer #2: Yes

5. Is the manuscript presented in an intelligible fashion and written in standard English?

Reviewer #1: Yes

Reviewer #2: Yes

6. Review Comments to the Author

Reviewer #1: (No Response)

Reviewer #2: (No Response)

7. PLOS authors have the option to publish the peer review history of their article (what does this mean?). If published, this will include your full peer review and any attached files.

Reviewer #1: No

Reviewer #2: **Yes: **Salwa mahmoud

---

## [Editor Report · Acceptance letter]

5 Oct 2021

PONE-D-21-23190R1 

Effect of hearing aids on body balance function in non-reverberant condition: A posturographic study 

Dear Dr. Hiraumi:

I'm pleased to inform you that your manuscript has been deemed suitable for publication in PLOS ONE. Congratulations! Your manuscript is now with our production department. 

Kind regards, 

on behalf of

Dr. Manabu Sakakibara 

Academic Editor

PLOS ONE